# The Difference in the Proportions of Deleterious Variations within and between Populations Influences the Estimation of FST

**DOI:** 10.3390/genes13020194

**Published:** 2022-01-22

**Authors:** Sankar Subramanian

**Affiliations:** GeneCology Research Centre, The University of the Sunshine Coast, 90 Sippy Downs Drive, Sippy Downs, QLD 4556, Australia; ssankara@usc.edu.au; Tel.: 61-7-5430-2873

**Keywords:** population differentiation, FST, deleterious mutations, temporal distributions and population genetics theory

## Abstract

Estimating the extent of genetic differentiation between populations is an important measure in population genetics, ecology and evolutionary biology. The fixation index, or FST, is an important measure, which is routinely used to quantify this. Previous studies have shown that the FST estimated for selectively constrained regions was significantly lower than that estimated for neutral regions. By deriving the theoretical relationship between FST at neutral and constrained sites, we show that excess in the fraction of deleterious variations segregating within populations compared to those segregating between populations is the cause for the reduction in FST estimated at constrained sites. Using whole-genome data, our results revealed that the magnitude of reduction in FST estimates obtained for selectively constrained regions was much higher for distantly related populations compared to those estimated for closely related pairs. For example, the reduction was 47% for comparison between Europeans and Africans, 30% for the European and Asian comparison, 16% for the Northern and Southern European pair, and only 4% for the comparison involving two Southern European (Italian and Spanish) populations. Since deleterious variants are purged over time due to purifying selection, their contribution to the among-population diversity at constrained sites decreases with the increase in the divergence between populations. However, within-population diversities remain the same for all pairs compared; therefore, the FST estimated at constrained sites for distantly related populations are much smaller than those estimated for closely related populations. We obtained similar results when only the SNPs with similar allele frequencies at neutral and constrained sites were used. Our results suggest that the level of population divergence should be considered when comparing constrained site FST estimates from different pairs of populations.

## 1. Introduction

Since the introduction of F-statistics by Sewall Wright [1], the fixation index, or *F*_ST_, has been routinely used to measure the extent of differentiation between populations [2,3,4,5,6,7,8,9,10,11,12]. *F*_ST_ compares the heterozygosities within and between (or total) populations to measure the level of genetic structure among populations. A number of methods have been developed to measure *F*_ST_ using genetic data, such as by [13] Nei (1973), [14] Weir and Cockerham (1984), and [15] Hudson (1992), which were based on Wright’s F-statistics. An alternative method based on genetic distances was developed to measure population differentiation in gene, gametic, and genotypic frequency data [16]. Furthermore, [17] Jost (2008) introduced another measure of differentiation, D, which measures the fraction of allelic variation among populations. Additionally, model-based Bayesian approaches [4] and moment estimators for measuring population-specific FST have been developed [18,19]. Recently, a novel method based on allele frequency difference (AFD) was also developed to measure population differentiation [20]. Despite the availability of many methods, the first three methods mentioned above are widely used in population genetics and evolutionary biology.

Apart from being an integral part of the descriptive statistics to describe a population, *F*_ST_ has direct applications in conservation biology, ecology, evolutionary biology, and clinical genetics. *F*_ST_ reveals the extent of genetic drift and the level of migrations between populations, which is useful to understand the population dynamics of an ecosystem [21]. The level of differentiation in populations helps conservation biologists to measure the risk of extinction of a population or species [22]. *F*_ST_ is also used to identify candidate genetic variants and genes associated with Mendelian and complex genetic diseases [2,3,9]. Furthermore, *F*_ST_ is used to infer genetic connectivity among populations [23], geographical patterns of deleterious mutations [24], and to prioritize SNPs for genomic selection studies (CHANG et al. 2019).

In evolutionary biology, *F*_ST_ is used to detect the signature of positive selection [3,4,6,7,10,11,12,25]. However, only a handful of studies examined the influence of selective constraints on *F*_ST_. A previous study reported lower *F*_ST_ for coding compared to noncoding SNPs [3]. The reduction in *F*_ST_ was more pronounced when only the amino acid changing nonsynonymous SNPs (nSNPs) were considered, and a similar reduction was observed for mutations in disease-related genes. This suggests that purifying selection does not allow an increase in the frequency of potentially deleterious nSNPs, which could have led to the observed low *F*_ST_ [26]. Later, a more systematic investigation was conducted to examine this issue using human genome data [27]. This study grouped nSNPs based on the evolutionary rates of sites in which they were present and showed a positive correlation between the rates and *F*_ST_. Hence, the *F*_ST_ estimated for the nSNPs present in selectively constrained sites (with a low rate of evolution) was much smaller than that estimated for those present in neutral sites with high evolutionary rates. A similar observation was made by another study on the populations of fruit flies from France and Rwanda [28]. *F*_ST_ estimates obtained for long introns (known to be under high purifying selection) and conserved genes were typically lower than those estimated for short introns (under relaxed selective constraints) and less conserved genes.

Although the influence of selective constraints on *F*_ST_ estimates has been documented, how exactly this affects *F*_ST_ estimations or the mechanism by which selective constraints influence these estimates is unclear. Furthermore, whether the magnitude of reduction in *F*_ST_ is dependent on the divergence between populations is unknown. To examine these, we first investigated the theoretical relationship between *F*_ST_ at neutral and constrained sites. Using data from the 1000 Genomes Project—Phase 3 [29], we then estimated *F*_ST_ for pairs of populations with different levels of divergence, such as Europeans–Africans, Europeans–Asians, Northern–Southern Europeans, and two Southern European populations (Italians and Spanish). We found that the difference in the *F*_ST_ estimated between the neutral and constrained sites was much higher for distantly related populations compared to the closely related population pair.

## 2. Materials and Methods

### 2.1. Estimating the Excess Fraction of Deleterious Variants Present within Population

Heterozygosity in neutral and selectively constrained sites can be expressed as follows:(1) Heterozygosity at synonymous (neutral) sites=H  
(2)Heterozygosity  at nonsynonymous (constrained) sites=Hf

The fraction of segregating mutations in the population is denoted as *f*, and this includes neutral and deleterious mutations (assuming the fraction of adaptive mutations is negligible). Therefore, it is the ratio of heterozygosity at constrained (*Hf*) and neutral (*H*) sites. This is similar to the notation *f*_0_ used by [30] Kimura (1983) for the fraction of neutral mutations/substitutions expected for species-level comparison (long-term evolution). However, in populations, slightly deleterious mutations are also expected to segregate in addition to neutral mutations.

In terms of heterozygosity, *F*_ST_ at synonymous sites (*F_ST(S)_*) can be expressed using Hudson et al. [15] as follows:(3)$$\;\;\;\;\;\;\;\;\;\;F_{ST\left(S\right)}=\frac{H_{b}-H_{w}}{H_{b}}$$
where *H_b_* and *H_w_* are synonymous site heterozygosity for between and within populations.

Using Equation (2), *F*_ST_ at nonsynonymous sites (*F_ST(N)_*) is given as
(4)$$F_{ST\left(N\right)}=\frac{H_{b}f_{b}-H_{w}f_{w}}{H_{b}f_{b}}$$
where *f_b_* and *f_w_* are fractions of neutral plus slightly deleterious nonsynonymous mutations segregating between and within populations, respectively. For comparisons involving two populations, $f_{w}=(f_{1}+f_{2})/2$, where *f*_1_ and *f*_2_ are the neutral + deleterious fractions in Populations 1 and 2, respectively. If the *f_b_* and *f_w_* fractions of nonsynonymous mutations are equal, then we can show that *F*_ST_ at synonymous sites is equal to that at nonsynonymous sites, as given below.
(5)$$\;\frac{H_{b}-H_{w}}{H_{b}}=\frac{H_{b}f_{b}-H_{w}f_{w}}{H_{b}f_{b}}\;\;\;\;\;\;\;\;\;\;\;\;\;\;if\;\;f_{b}=f_{w}$$
(6)$$F_{ST\left(S\right)}=F_{ST\left(N\right)}\;\;\;\;\;\;\;\;\;\;\;\;\;\;\;\;\;\;\;\;\;\;\;\;\;\;if\;\;f_{b}=f_{w}$$

However, it is well known that the fraction of slightly deleterious mutations segregating within a population is higher than that segregates between populations. This is because a much higher fraction of those segregating within population are young and yet to be purged from the population by natural selection. Therefore, *f_w_* is expected to be higher than *f_b_*. Hence, we get
(7)$$\frac{H_{b}-H_{w}}{H_{b}}>\frac{H_{b}f_{b}-H_{w}f_{w}}{H_{b}f_{b}}\;\;\;\;\;\;\;\;\;\;\;\;\;\;\;if\;\;f_{b}<f_{w}$$

The above equation could be simplified by converting the fraction *f_w_* in terms of *f_b_* as
(8)$$f_{w}=f_{b}\left(1+\eta \right)\;$$

In the above equation, *η* is the excess fraction of deleterious variations segregating within populations compared to those segregating between populations. Substituting this for *f_w_* we get
(9)$$\;\frac{H_{b}-H_{w}}{H_{b}}>\frac{H_{b}-H_{w}\left(1+\eta \right)}{H_{b}}\;\;\;\;\;\;\;\;if\;\;f_{b}<f_{w}\;or\;\eta >0\;$$
(10)$$F_{ST\left(S\right)}>F_{ST\left(N\right)}\;\;\;\;\;\;\;\;\;\;\;\;\;\;if\;\;f_{b}<f_{w}\;or\;\eta >0\;$$

The above relationships clearly show that if *f_w_* is higher than *f_b_* or if there is an excess in the proportion of deleterious variations segregating within populations compared to that between populations (*η*), then the *F*_ST_ of the nonsynonymous sites will be smaller than that of synonymous sites. The magnitude of the reduction in the *F*_ST_ of the nonsynonymous sites could be quantified as
(11)$$\rho =1-\frac{F_{ST\left(N\right)}}{F_{ST\left(S\right)}}\;\;\;$$
which is
(12)$$\rho =1-\frac{H_{b}-H_{w}\left(1+\eta \right)}{H_{b}-H_{w}}$$

Equation (12) shows the theoretical relationship between *ρ* and *η*. However, using Equation (11), *ρ* can be empirically estimated for the exome data using the estimates of *F*_ST_ at neutral ($\tilde{F}$*_ST(S)_*) and constrained sites ($\tilde{F}$*_ST(N)_*).

A similar relationship for Nei’s *F*_ST_ (*G*_ST_) for neutral and constrained sites is
(13)$$\rho =1-\frac{H_{T}-H_{S}\left(1+\eta \right)}{H_{T}-H_{S}}$$
where *H_T_* and *H_S_* are the heterozygosity of the total and subpopulations.

### 2.2. Population Genome Data

Whole-genome data for 516 humans belonging to 5 worldwide populations were downloaded from the 1000 Genomes Project—Phase 3 (ftp://ftp.1000genomes.ebi.ac.uk/vol1/ftp/release/20130502/, accessed on 25 May 2019) [29]. This includes British (91), Han Chinese (103), Italian (107), Spanish (107), and Nigerian/Yoruban (108) populations. Only biallelic single nucleotide polymorphisms (SNPs) from the autosomes were included in the analyses. The allele frequencies of SNPs were computed separately for each population. These were then used for estimating *F*_ST_ using the estimators described below. Pairwise FSTs were computed for the exomes of TSI (Italian)–YRI (Nigerian), TSI (Italian)–CHB (Chinese), TSI (Italian)–GBR (British), and TSI (Italian)–IBS (Spanish). In population genetics, the ratio of synonymous-to-nonsynonymous divergence/diversity is used to measure the magnitude of selective constraints. However, here we used the Combined Annotation-Dependent Depletion (CADD) [31] for this purpose. To measure the level of selective constraints on a site, this method uses the information from sequence conservation, properties of amino acid changes, allele frequency, protein structural motif, transcription regulation signals, chromatin structure, and include the scores from already established methods such as GERP, PhyloP, PolyPhen, SIFT, and Grantham. This robust method integrates these diverse annotations into a single measure (*C* score). The precomputed C scores for the 1000 Genomes Project data are available at http://cadd.gs.washington.edu/download/ (accessed on 25 May 2019), and these scores were mapped to the genotype data from the 1000 genome project. To identify the derived alleles, orientations of SNVs were determined using the ancestral state of the nucleotides, which was inferred from six primate EPO alignments [29]. 

### 2.3. F_ST_ Estimation

For estimating *F*_ST_ from human exome data, we used two methods developed by Hudson et al. [15] and Nei [13]. We used the following estimators:$${\tilde{F}}_{ST}^{Hudson}=\frac{H_{B}-H_{S}}{H_{B}}$$
$${\tilde{F}}_{ST}^{Nei}=\frac{H_{T}-H_{S}}{H_{T}}$$
and
$$H_{T}=2\frac{{\tilde{p}}_{1}+{\tilde{p}}_{2}}{2}\left(1-\frac{{\tilde{p}}_{1}+{\tilde{p}}_{2}}{2}\right)$$
$$\;H_{S}={\tilde{p}}_{1}\left(1-{\tilde{p}}_{1}\right)+{\tilde{p}}_{2}\left(1-{\tilde{p}}_{2}\right)$$
$$\;H_{B}={\tilde{p}}_{1}\left(1-{\tilde{p}}_{2}\right)+{\tilde{p}}_{2}\left(1-{\tilde{p}}_{1}\right)$$
where *p*_1_, *p*_2_ are frequencies of the two populations. To combine the *F*_ST_ estimated for the different SNPs of the genome, we used the ratio of averages approach suggested by Bhatia et al. [32]. This was done by calculating the averages of the numerator and denominator of the equations separately and the ratio of these was computed. To estimate the variance, we used a bootstrap resampling procedure with 1000 replicates. The SNPs were sampled with replacement, and 1000 pseudoreplicates were generated. This was then used to estimate the variance. To determine whether the *F*_ST_ estimated for neutral synonymous sites was significantly higher than that obtained for the conserved nonsynonymous sites, the *Z*-test was used.

## 3. Results

### 3.1. The Effect of Purifying Selection on F_ST_

To examine the influence of selective constraints on *F*_ST_, we used European and African exome data from the 1000 Genomes Project—Phase 3 (see Materials and Methods). In order to examine the magnitude of selection pressure, the Combined Annotation Dependent Depletion (CADD) score, or *C*-score, was used [31]. Nonsynonymous SNPs were grouped into seven categories based on their *C*-scores. Figure 1A shows the relationship between selection pressure and *F*_ST_ estimated for synonymous (sSNPs) and nonsynonymous SNPs (nSNPs) using the exome data for the Italian (TSI)–Nigerian (YRI) pair. Clearly, *F*_ST_ is the highest for the neutral sSNPs, which declines with an increase in selection.

The *F*_ST_ estimate for highly constrained nSNPs with a *C*-score > 30 was only 0.082, which is much smaller than that estimated for sSNPs (0.154). We introduced a measure, *ρ*, to capture the magnitude of the reduction in *F*_ST_ estimates of nSNPs compared to that of sSNPs (Equation (11), see Materials and Methods). Figure 1B shows the positive relationship between the extent of the selective constraint and the magnitude of the reduction in *F*_ST_ (*ρ*). The reduction in the *F*_ST_ estimate was only 2% for nSNPs under a relaxed selection pressure (*C*-Score ≤ 5), which increases with the magnitude in selection pressure. For highly constrained nSNPs (*C*-score > 30), the reduction in *F*_ST_ was 47%, which is 24 times higher than that observed for nSNPs under a relaxed constraint. Please note that the results shown were based on Hudson’s estimator and the results obtained using Nei’s estimator are given in the Appendix A.

### 3.2. Relationship between the F_ST_ Values at the Neutral and Constrained Genomic Regions

To understand the actual cause of the reduction in *F*_ST_ for constrained SNPs, we examined the theoretical relationship between *F*_ST_ at neutral and constrained regions. We showed that the fractions of neutral + deleterious variations segregating between (*f_b_*) and within (*f_w_*) populations hold the answer to this. If these fractions were similar (*f_b_* = *f_w_*), then the FST estimates for sSNPs and nSNPs are expected to be equal (Equation (5)). However, it is well known that a higher proportion of slightly deleterious SNPs is expected to segregate within populations rather than between populations. This is because a significant fraction of them are purged by purifying selection over time, and hence their fraction gets diminished for between-population comparisons. Therefore, we show that the *F*_ST_ estimated for nSNPs is expected to be smaller than that of sSNPs as the fraction of neutral + deleterious SNPs segregating within populations is higher than those segregating between populations (*f_w_* > *f_b_*) (Equation (7)). To quantify the magnitude of difference between the two fractions, we proposed the measure *η*, which is the excess fraction of deleterious SNPs segregating within populations than those present between populations (Equations (8) and (9)). We show the relationship between *η* and the magnitude of the reduction in *F*_ST_ estimated for nSNPs compared to that of sSNPs (*ρ*) (Equation (12)), which clearly shows that *ρ* is dependent on *η*.

Using the within (*H*_w_) and between (*H*_b_) population heterozygosities for the sSNPs of the European–African comparison, the theoretical relationship between *ρ* and *η* (based on Equation (12)) was plotted. Figure 2 reveals a positive correlation between the two variables. The values of *ρ* estimated for the nSNPs (using Equation (11)) belonging to the seven selective constraint categories (*C*-scores) were overlaid on the theoretical line, and the corresponding *η* values were predicted (red dots on the line). This suggests that for highly constrained SNPs (*C*-score > 30) there is an 8.6% excess fraction of deleterious SNPs present within populations compared to those segregating between populations, and this results in a 47% reduction in the *F*_ST_ estimate. This excess was only 0.3% for the SNPs under relaxed selective constraints (*C*-score ≤ 5), which resulted in a 2% reduction in the *F*_ST_. Hence, these results suggest that the magnitude of reduction is indeed dictated by the excess fraction of deleterious SNPs segregating within populations compared to those segregating between populations.

### 3.3. F_ST_ Estimates and Population Divergence

Next, we investigated the effects of selection constraints on *F*_ST_ with respect to population divergence. This is to compare the magnitude of the reduction in *F*_ST_ estimated for closely and distantly related populations. For this purpose, we used four pairs of comparisons with different levels of divergence, European (Italian/TSI)–African (Nigerian/YRI), European (Italian/TSI)–Asian (Chinese/CHB), Southern European (Italian/TSI)–Northern European (British/GBR), and two Southern Europeans (Italian/TSI–Spanish/IBS). Figure 3A–D shows the *F*_ST_ estimates obtained for sSNPs and highly constrained nSNPs for the four pairs of populations. This pattern suggests that there is a positive correlation between the population divergence and the extent of the reduction in *F*_ST_, which is clear in Figure 4. The *F*_ST_ observed for constrained nSNPs of the distantly related Italian–Nigerian pair was 47% smaller than that of the sSNPs (Figure 4). While this reduction was 30% for the Italian–Chinese pair and 16% for Italian–British comparison, it was only 4% for the closely related Italian–Spanish pair (Figure 4).

We then examined the theoretical relationship by plotting the relationship between *ρ* and *η* (Equation (12)) for the four pairs populations. For this purpose, we used the within- and between-population heterozygosity estimates of the sSNPs of the Italian–Nigerian, Italian–Chinese, Italian–British, and Italian–Spanish populations. While all four relationships show a positive trend between *ρ* and *η*, there was a huge difference in the slopes of these relationships (Figure 5). The slopes observed for the closely related pairs are much higher than that of the distantly related pair. Using the expected theoretical lines, the corresponding *η* values were predicted for the *ρ* values estimated for the four pairs of populations (red dots on the lines). This analysis showed that the excess fractions of the 8.6%, 4.0%, 0.2%, and 0.03% slightly deleterious nSNPs are present within populations rather than between populations of the Italian–Nigerian, Italian–Chinese, Italian–British, and Italian–Spanish pairs, respectively. The presence of these excess fractions resulted in a 47%, 30%, 16%, and 4% reduction in the *F*_ST_ estimated for the highly constrained nSNPs (*C*-score > 30) of the corresponding pairs of populations, respectively.

## 4. Discussion

Although previous studies have observed a reduction in the *F*_ST_ estimates of selectively constrained sites [3,27,28], the true cause for that reduction was established in this study. Using the theoretical relationship between *F*_ST_ at neutral and constrained sites, we showed that an excess fraction of deleterious mutations segregating within population compared to those between populations (*η*) is the reason for the reduction in *F*_ST_ at the constrained sites. We also showed the relationship between *η* and the magnitude of the reduction in the *F*_ST_ of constrained nSNPs in comparison to that of neutral sSNPs (*ρ*). The reason for the excess fraction *η* present within populations is due to the fact that a high proportion of deleterious mutations segregating within populations are relatively young and hence were not removed by natural selection. Therefore, they contribute significantly to the constrained site heterozygosity within populations. In contrast, a much higher proportion of the harmful mutations have been purged due to the time elapsed, and hence their contribution to the constrained site heterozygosity between populations is relatively less. Hence, within population heterozygosity at constrained sites is much more inflated than that observed for the inter-population comparison. This results in the reduction of the *F*_ST_ estimates, as it is based on the normalized difference between the inter- and intra-population diversities.

The results of this study highlight two important patterns and provide theoretical and empirical explanations for them. First, the reduction in the *F*_ST_ estimates positively correlates with the magnitude of selection, suggesting a much higher underestimation for nSNPs at highly constrained regions of the genome. This is because the high magnitude of the selective constraints leads to segregation of more slightly deleterious mutations within populations (as more genomic sites are under selection) and hence the fraction of deleterious nSNPs segregating within populations will be much higher than those segregating between populations (*f_w_* >> *f_b_* or *η* >> 0). Hence, this leads to a much higher reduction in the *F*_ST_ of nSNPs at the highly constrained regions compared to that of sSNPs (*F*_ST(N)_ << *F*_ST(S)_). In contrast, there are fewer deleterious nSNPs in the less constrained regions and hence the fraction of harmful polymorphisms segregating within populations is expected to be only modestly higher than those segregating between populations (*f*_w_ > *f*_b_ or *η* > 0). This results in a much smaller reduction in the *F*_ST_ estimated for nSNPs present in regions under relaxed selective constraints (*F_ST(N_*_)_ < *F_ST(S)_*).

Second, we have shown that the magnitude of the reduction in *F*_ST_ at the constrained sites for comparisons involving distantly related populations was much higher than that of those involving closely related pairs. For instance, this reduction for the European–African comparison (47%) is more than ten-fold higher than that of the Southern European pair (Italian–Spanish) (4.2%). It is well known that deleterious variants are removed over time and hence the only a small fraction (*f*_b_ << 1) of them segregate and contribute to constrained site inter-population diversity for distantly related populations. However, a relatively modest fraction (*f_b_* < 1) of harmful nSNPs contribute to the inter-population diversity for the closely related population as the elapsed time is not enough to purge most of them. On the other hand, the fraction of deleterious nSNPs within a population (*f_w_*) remains the same for comparisons involving both distantly as well as closely related populations. The excess fraction *η* (which is the normalized difference between *f_w_* and *f_b_*) is much smaller for the comparisons involving closely related populations (*η* << 1) than those involving distantly related populations (*η* < 1). Hence, the magnitude of the reduction in the constrained site *F*_ST_ (with respect to neutral site *F*_ST_) for distantly related populations (e.g., European–African) is much higher (*F_ST(N)_* << *F_ST(S_*_)_) than that observed for closely related populations (*F_ST(N)_* < *F_ST(S)_*) (e.g., Italian–Spanish).

In this study, we used the formula of Hudson et al. [15] to derive the relationship between *F*_ST_ at neutral and constrained sites and also to estimate *F*_ST_ from exome data. This method compares heterozygosities between and within populations. In contrast, Nei [13] developed a method that compares heterozygosities of the total and subpopulations. Therefore, we derived the relationship between *F*_ST_ at neutral and constrained sites for the method of Nei as well (Equation (13)) and repeated all analyses using Nei’s estimator (Appendix A). However, this analysis produced similar results to those obtained using the method of Hudson et al.

It is well known that the allele frequencies of constrained nSNPs are typically lower than those of neutral sSNPs. Therefore, the difference in the allele frequency alone could bias the estimation of the *F*_ST_ of these SNPs. Hence, we included only the rare sSNPs and nSNPs with an allele frequency <0.5% and computed the *F_ST(S)_* and *F_ST(N)_* for the four population pairs. The mean allele frequency of the sSNPs and nSNPs were comparable for each pair of populations. These values are 0.25% and 0.28% for the Italian–Nigerian comparison, 0.32% and 2.9% for the Italian–Chinese pair, 0.43% and 0.35% for the Italian–British, and 0.38% and 0.32% for the Italian–Spanish comparisons. For this dataset, the magnitude of the reduction in the *F*_ST_ of the nSNPs (compared to sSNPs) were 24%, 19%, 6%, and 2% for the Italian–Nigerian, Italian–Chinese, Italian–British, and Italian–Spanish comparisons, respectively.

The findings of this study suggest that the *F*_ST_ estimated for different genes or genomic regions of a genome are not comparable if the level of the selective constraints is different between them. This is particularly important when using *F*_ST_ estimates to detect positive selection because such methods assume neutral evolution in genes and genomic regions and hence do not account for excess deleterious mutations that have not been purged out from the populations [4,6,10,11,12,25]. Our results also strongly indicate that the *F*_ST_ obtained from the constrained regions of different pairs of populations are not comparable if the population divergence times between the pairs are not the same. In such cases, *F*_ST_ estimations should include only neutral sites to obtain unbiased estimates. However, this is only possible for large genomes such as vertebrates in which constrained regions constitute only a small fraction (~10%) of the genome [33,34]. This is an important issue for small genomes such as those of fruit flies where >50% of the genome is under selection [35]. Therefore, population divergence time needs to be considered when comparing the genome-wide *F*_ST_ estimates between different populations.

## Figures and Tables

**Figure 1 genes-13-00194-f001:**
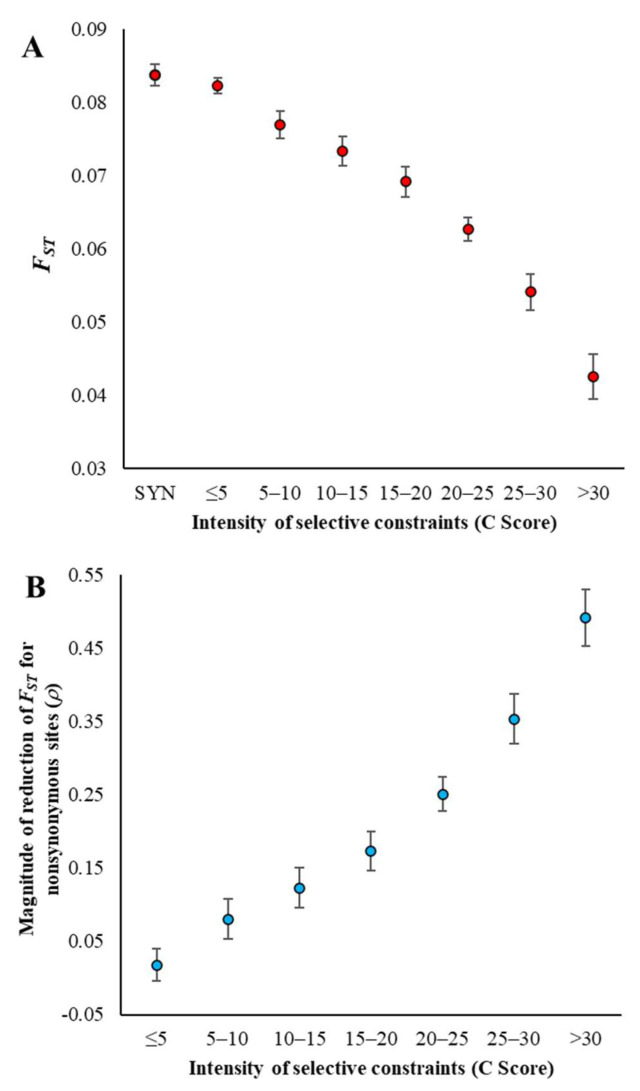
(**A**) Relationship between selection intensity and *F*_ST_. Whole-exome data comprising synonymous SNPs (sSNPs) and nonsynonymous SNPs (nSNPs) for the Italian (TSI)–Nigerian (YRI) population pair was used to estimate *F*_ST_. The magnitude of selection intensity on nSNPs was measured by the Combined Annotation-Dependent Depletion (CADD) method that integrates many diverse annotations into a single measure (*C* score) [31]. A bootstrap resampling procedure (1000 replicates) was used to estimate the standard error. (**B**) The magnitude of the reduction in the *F*_ST_ estimates and selection intensity. The X-axis shows the reduction in *F*_ST_ estimates of nSNPs in comparison with that of sSNPs (*ρ*) using Equation (11) (see Materials and Methods) for the exome data described above. Error bars show the standard error of the mean.

**Figure 2 genes-13-00194-f002:**
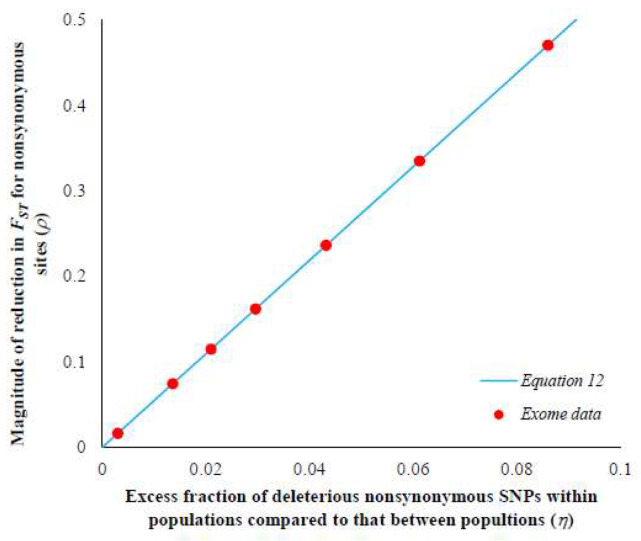
The theoretical relationship between the excess in the fraction of deleterious mutations segregating within population compared to between populations (*η*) and the magnitude of the reduction in the *F*_ST_ estimates of nSNPs (*ρ*) using Equation (12) (see Materials and Methods). The line was plotted using within- and between-population heterozygosities of neutral sSNPs for the Italian–Nigerian comparison and the red dots are the *ρ* values estimated from the exome data using Equation (11). Using the theoretical expected line, *η* values were predicted for the corresponding observed *ρ* values.

**Figure 3 genes-13-00194-f003:**
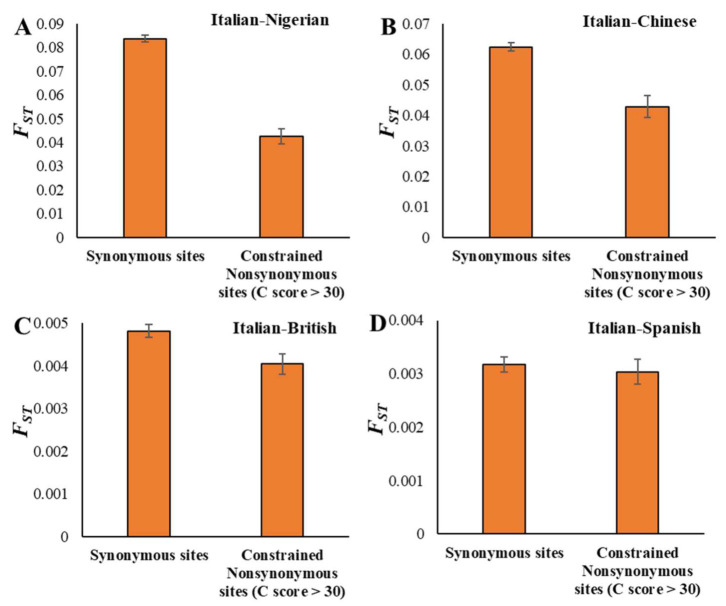
*F*_ST_ estimates for synonymous and highly constrained nonsynonymous SNPs of the (**A**) Italian–Nigerian, (**B**) Italian–Chinese, (**C**) Italian–British, and (**D**) Italian–Spanish population pairs. Error bars are the standard error of the mean, and a bootstrap resampling procedure (1000 replicates) was used to estimate the variance. The difference between the FST estimates of the neutral and constrained sites are highly significant (*p* < 0.01, *Z* test) for three comparisons and not significant for the Italian–Spanish pair.

**Figure 4 genes-13-00194-f004:**
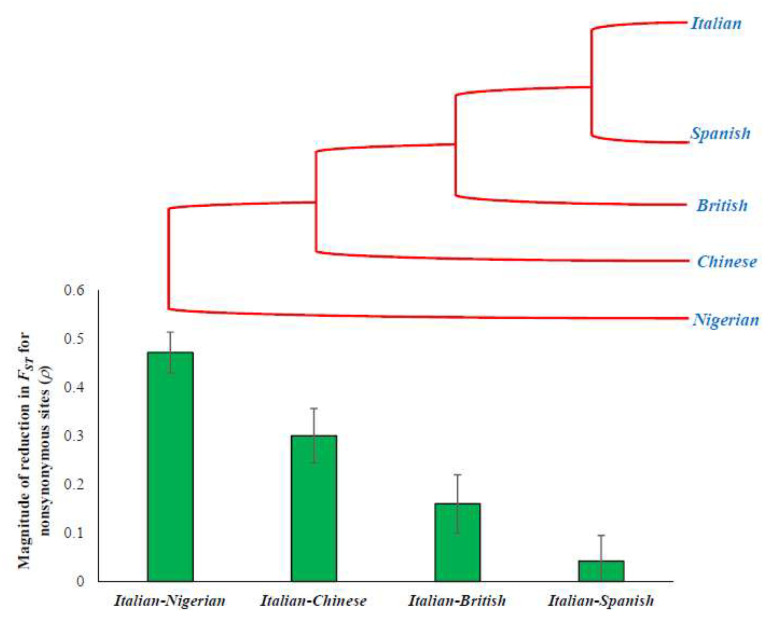
The magnitude of the reduction in *F*_ST_ estimates of the nSNPs obtained for four population pairs. The population tree on top is drawn to highlight the correlation between the population divergence and the magnitude of the reduction in *F*_ST_.

**Figure 5 genes-13-00194-f005:**
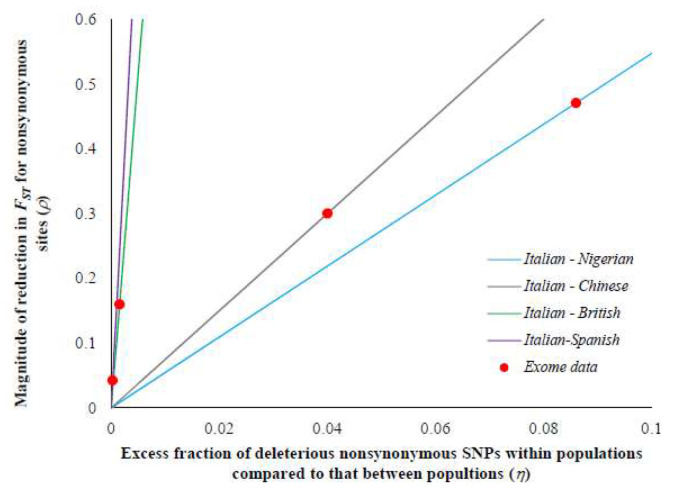
The theoretical relationship between the excess in the fraction of deleterious mutations segregating within population and those between populations (*η*) and the magnitude of the reduction in the *F*_ST_ estimates of nSNPs (*ρ*) using Equation (12) (see Materials and Methods). Neutral population diversities based on sSNPs for within- and between-population comparisons of the Italian–Nigerian, Italian–Chinese, Italian–British, and Italian–Spanish pairs were used to plot the lines, and the *ρ* estimated from the exome data (using Equation (11)) are shown as red closed circles. The theoretical lines were used to predict the *η* values of the corresponding *ρ* estimated using the exome data.

## Data Availability

Not applicable.

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
