# Peer review of "The Difference in the Proportions of Deleterious Variations within and between Populations Influences the Estimation of FST"

_genes, 2022, doi:10.3390/genes13020194_

Round 1

Reviewer 1 Report

In this manuscript, the author demonstrated the positive relationship of Fst reduction magnitude (Fst(n) – Fst(s)) and intensity of selective constrains (based on categories of C-score). Based on theoretical (??=??(1+?) and other equation) deduction, the author suggested that the much higher magnitude of Fst reduction within highly constrained gene regions is led by the higher excess fraction of deleterious mutations within populations compared to that between populations. A fundamental theory/hypothesis in this paper is that the fraction of slightly deleterious mutations fixed within population (closely related) is higher than that between populations (distantly related). Although the impact of conservation of genes in Fst estimates has been documented in previous publications, the author answers the questions how and why. This study is very helpful for the researchers working with population datasets to design analysis in proper way. However, there are concerns that need to be addressed, as detailed below.

  1. In Introduction, it seems that the state-of-art in this research field is not systematically described. Some more recent studies are not included (between 2015-2020).

  1. One major point about this paper is the statement of purifying selection (negative selection) and the use of this term in multiple places of the manuscript. It seems that that “selective constraint” is equivalent to “purifying selection” as they are used in the same way in the manuscript. For example:
  • Line 52-53, does the author actually mean “the influence of selective constraints on Fst?” Or “the influence of deleterious mutations on Fst?” Fst is used to detect positive selection in these studies (line 47-58). Likewise, people can use Fst to detect the extent of purifying selection as well. This study is trying to suggest that people should be cautious to make an inference of genetic differentiation between populations based on Fst as the estimate is biased when the populations are distantly related.
  • Same case as above point in line 70-72. “Although the influence of selective constrains on Fst estimates has been...” this makes more sense for me.
  • In Figure 1, the x-axis is “Intensity of purifying selection” (C Score). I understand that the magnitude of selection intensity on nSNPs (or selective constrains) could be indicated by C score. People normally estimate purifying selection using dN/dS. (Regarding C score, there is a question for it below).
  • Line 231, and line 297.
  • Line 335-336, would it be better to change into something like “…such methods assume neutral evolution in genes...hence do not account for excess deleterious mutations that have not been purged out from the populations.”?

  1. In Method, it would be good to explain more in using C score instead of dN/dS for the readers.

  1. Line 88-89, please add the citation reference and explain f more precisely.

  1. Line 133, the author mentioned that 26 populations were used. However, only five population data was used in the study. Sample size affect the estimates of H, and even stronger in constrained sites (in the author’s previous study and other studies). How many samples did the author use for each population in the analysis? Have you considered the influence and if it has anything to do with your results?

  1. Line 136, how to calculate the allele frequencies of SNP? SNP occurrence across the whole data selected in the analysis? Or else?

  1. Line 157-159, It would be nice that more details are described for this method part.

  1. Figure 3, the layout of the figure can be improved, and the main title should be labelled in the figure panels themselves (name of the comparison pairs).

  1. Line 262, could the author explain why Z test is used here?

Minor points:

  1. Line 55, what does the author mean genic and nongenic SNPs? Coding genes and non-coding genes?

  1. Line 58-59, Or say “purifying selection does not allow an increase in the frequency of deleterious nSNPs (or mutations)”?

  1. Line 63-64, I feel that it is meaningless to compare Fst between groups with low and high evolutionary rates as this is the criterion of this study selecting data (line 61-62). The comparison makes more sense to me like, 1) calculate Fst between constrained genes and neutral genes when all sites are considered, or sSNPs and nSNPs are considered separately. 2) compare the difference of Fst changes among total SNPs, sSNPs and nSNPs, just like what the author was doing in the simulation.

  1. Line 72-74, basically, these two questions are the same question.

  1. In the end of the Introduction, it would be nice to add one more sentence about the main conclusion.

  1. Line 144, the data is not available yet.

  1. Line 197-206, a bit repeated content as that in Method.

  1. Line 301, “underestimation”. Would it be better to change as “reduction” or “decreasing”?

  1. Line 319-321, “Therefor… involving distantly related populations”, it is not a consequence relationship, but that in line 321-324 is.

Author Response

Response to Reviewer 1 comments

Comment 1:  In Introduction, it seems that the state-of-art in this research field is not systematically described. Some more recent studies are not included (between 2015-2020).

Response 1: We have included more recent studies (between 2017-2021) in the current version of the manuscript (lines 39-62) and also added the relevant references.

Comment 2:  One major point about this paper is the statement of purifying selection (negative selection) and the use of this term in multiple places of the manuscript. It seems that that “selective constraint” is equivalent to “purifying selection” as they are used in the same way in the manuscript. For example:

  • Line 52-53, does the author actually mean “the influence of selective constraints on Fst?” Or “the influence of deleterious mutations on Fst?” Fst is used to detect positive selection in these studies (line 47-58). Likewise, people can use Fst to detect the extent of purifying selection as well. This study is trying to suggest that people should be cautious to make an inference of genetic differentiation between populations based on Fst as the estimate is biased when the populations are distantly related.
  • Same case as above point in line 70-72. “Although the influence of selective constrains on Fst estimates has been...” this makes more sense for me.
  • In Figure 1, the x-axis is “Intensity of purifying selection” (C Score). I understand that the magnitude of selection intensity on nSNPs (or selective constrains) could be indicated by C score. People normally estimate purifying selection using dN/dS. (Regarding C score, there is a question for it below).
  • Line 231, and line 297.
  • Line 335-336, would it be better to change into something like “…such methods assume neutral evolution in genes...hence do not account for excess deleterious mutations that have not been purged out from the populations.”?

Response 2:  We agree and used the term ‘selective constraints’ and corrected the text in the lines mentioned above (see: lines 65-66, 83-85, 260, 332, 382-383).  The X-axis of Figure 1 has also been modified as suggested.

Comment 3:  In Method, it would be good to explain more in using C score instead of dN/dS for the readers.

Response 3:  Agreed.  We have discussed about use of C score over dN/dS in lines 159-166.

Comment 4:  Line 88-89, please add the citation reference and explain f more precisely.

Response 4:  We have clarified f more precisely and added the reference (lines 100 – 106)

Comment 5:  Line 133, the author mentioned that 26 populations were used. However, only five population data was used in the study. Sample size affect the estimates of H, and even stronger in constrained sites (in the author’s previous study and other studies). How many samples did the author use for each population in the analysis? Have you considered the influence and if it has anything to do with your results?

Response 5:  We have corrected this as we only used five populations for this study.  Many thanks for the reviewer to pointing out this error.  However, we used 91-108 individuals (516 in total) for each population.  Therefore, the sample size quite high and comparable between populations and hence we believe that there is no bias in estimating FST.  We have now mentioned the sample sizes and populations clearly in lines 150-154.

Comment 6:  Line 136, how to calculate the allele frequencies of SNP? SNP occurrence across the whole data selected in the analysis? Or else?

Response 6:  Calculation of allele frequencies was done for each population separately.  This has been clearly documented now in lines 155-156.

Comment 7:  Line 157-159, It would be nice that more details are described for this method part.

Response 7: More description about the method has been included (lines 184-188).

Comment 8:  Figure 3, the layout of the figure can be improved, and the main title should be labelled in the figure panels themselves (name of the comparison pairs).

Response 8:  The layout of Figure 3 has been modified and the labels were included in the figure panels.

Comment 9:  Line 262, could the author explain why Z test is used here?

Response 9:  This has now been explained in the methods section (lines 188–190).

Comment 10:  Line 55, what does the author mean genic and nongenic SNPs? Coding genes and non-coding genes?

Response 10: Yes.  This was amended (line 67).

Comment 11:  Line 58-59, Or say “purifying selection does not allow an increase in the frequency of deleterious nSNPs (or mutations)”?

Response 11: Done (lines 70-72).

Comment 12:  Line 63-64, I feel that it is meaningless to compare Fst between groups with low and high evolutionary rates as this is the criterion of this study selecting data (line 61-62). The comparison makes more sense to me like, 1) calculate Fst between constrained genes and neutral genes when all sites are considered, or sSNPs and nSNPs are considered separately. 2) compare the difference of Fst changes among total SNPs, sSNPs and nSNPs, just like what the author was doing in the simulation.

Response 12:  We agree with the reviewer as it is not the right way of comparing Fst estimated for regions with low and high evolutionary rates.  However, this was done by a previous study, and we simply reported that in our paper.

Comment 13:  Line 72-74, basically, these two questions are the same question.

Response 13:  We agree and now corrected this (lines 85-86).

Comment 14:  In the end of the Introduction, it would be nice to add one more sentence about the main conclusion.

Response 14:  We have added the main conclusion at the end of the introduction section (lines 91-93).

Comment 15:  Line 144, the data is not available yet.

Response 15:  The data is available now.  We have corrected the web link. (lines 167-168).

Comment 16:  Line 197-206, a bit repeated content as that in Method.

Response 16:  We agree.  But we want to reiterate this in the result section as well to make it clear to non-specialist readers

Comment 17:  Line 301, “underestimation”. Would it be better to change as “reduction” or “decreasing”?

Response 17: Corrected (line 335).

Comment 18:  Line 319-321, “Therefor… involving distantly related populations”, it is not a consequence relationship, but that in line 321-324 is.

Response 18:  This has been amended (lines 353-358).

Reviewer 2 Report

The paper adresses an interesting question, methods are adequate and conclusions are clearly written.

However, the Fst measure, although widely used, is not completely without problems. Thus, the paper should deal a bit more with the existing literature and maby take altenatives to Nei´s Fst into account. So, e.g. Gregorius & Roberds 1986 published an interesting paper. This and maybe some other publications should at least be discussed.

Author Response

Response to Reviewer 2 comments

Comment 1:  However, the Fst measure, although widely used, is not completely without problems. Thus, the paper should deal a bit more with the existing literature and maby take altenatives to Nei´s Fst into account. So, e.g. Gregorius & Roberds 1986 published an interesting paper. This and maybe some other publications should at least be discussed.

Response 1:  We agree.  As suggested by the reviewer, we have now included a detailed discussion about many other alternative methods including Gregorius and Roberds 1986 method and added several references (lines 39-50).